# MicroRNAs in T Cell-Immunotherapy

**DOI:** 10.3390/ijms24010250

**Published:** 2022-12-23

**Authors:** Sara G. Dosil, Ana Rodríguez-Galán, Francisco Sánchez-Madrid, Lola Fernández-Messina

**Affiliations:** 1Immunology Service, Hospital Universitario de la Princesa, Instituto Investigación Sanitaria Princesa, 28006 Madrid, Spain; 2Intercellular Communication in the Inflammatory Response, Vascular Pathophysiology Area, National Center for Cardiovascular Research (CNIC), 28029 Madrid, Spain; 3Universidad Autónoma de Madrid, 28049 Madrid, Spain; 4Centro de Investigación Biomédica en Red, Enfermedades Cardiovasculares (CIBERCV), 28029 Madrid, Spain; 5Department of Cell Biology, Faculty of Biological Sciences, Universidad Complutense de Madrid, 28040 Madrid, Spain

**Keywords:** microRNAs (miRNAs), antagomiRNAs (antagomiRs), immunotherapy, T cell immunotherapy, nanoparticles (NPs), nanomedicine, miRNA delivery

## Abstract

MicroRNAs (miRNAs) act as master regulators of gene expression in homeostasis and disease. Despite the rapidly growing body of evidence on the theranostic potential of restoring miRNA levels in pre-clinical models, the translation into clinics remains limited. Here, we review the current knowledge of miRNAs as T-cell targeting immunotherapeutic tools, and we offer an overview of the recent advances in miRNA delivery strategies, clinical trials and future perspectives in RNA interference technologies.

## 1. Introduction

Targeted delivery of RNA has attracted great interest in the last few years as a promising therapeutic strategy to modulate gene expression. This involves the administration of therapeutic exogenous nucleic acids, including messenger RNAs (mRNAs), and RNA interfering molecules, such as small interfering RNAs (siRNA), miRNAs, or antisense oligonucleotides (antagomiRs) [1]. In 2018, the first therapeutic approach using siRNAs was approved by the United States Food and Drug Administration (FDA) to treat Duchenne muscular dystrophy in patients with a rare mutation [2]. Thereafter, RNA-based therapeutics have experienced a rapid development and it is worth mentioning that mRNA-based vaccines, coding for antigenic pathogen proteins to induce a specific host immunological response [3], have been crucial to controlling the recent SARS-CoV-2 outbreak. Indeed, mRNA vaccines were rapidly and efficiently designed and translated into clinics, showing an efficiency of around 95% in preventing COVID-19 disease, and providing a persistent immune protection [4].

miRNAs are endogenous small (~19–24 nucleotides) non-coding RNAs, capable of regulating gene expression [5]. A myriad of studies have identified the dysregulation of miRNAs in disease, highlighting their potential as biomarkers for diagnosis and prognosis in several pathologies, including cancer [6,7], cardiovascular diseases [8,9], or immune-related diseases, as discussed below, among others. In fact, very comprehensive articles have extensively reviewed the role of miRNAs in immune modulation [10,11]. Interestingly, a single miRNA can target different mRNA targets within complex regulatory networks, and have great potential to control immune function and inflammatory cellular pathways [12].

Despite their great possibilities, the use of miRNAs in human therapy is limited, mainly due to their biological low stability, their inefficient delivery to specific tissues, and their potential off-target effects [13]. Several strategies to avoid some of these drawbacks have been explored, including RNA modifications, the use of nanocarriers, extracellular vesicles, or viral-based delivery systems. Here, we provide a perspective of the recent advances in miRNA delivery therapeutics, with a special focus on their use as T cell immunoregulators in disease.

## 2. miRNAs as T Cell Immune Modulators

Efficient host detection of antigens triggers the recruitment, activation, and differentiation of T lymphocytes. These central players in cell-mediated immunity can be classified into two major subsets: CD4+ T helper (Th) lymphocytes, which modulate immune responses through the activation of other immune subsets and release of cytokines; and CD8+ T cytotoxic lymphocytes, which directly recognize and kill infected or transformed cells. Th1, Th2, Th17, follicular helper T cells, and regulatory T (Treg) cells are among the principal Th lineages. The balance of these effector populations plays a pivotal role in controlling pathogen clearance and tumor immune surveillance, while maintaining tissue homeostasis. Lineage commitment has been mainly linked to the strength of the interaction of the T cell receptor (TCR) with the antigen and to the presence of cytokines in the microenvironment [14,15]. Given the critical role of CD4+ T lymphocytes, their development, function and polarization are tightly regulated by transcription factors and post-transcriptional modulators, including miRNAs.

After antigen encounter, T cells undergo a genetic switch, promoting proliferation and effector signals. While many of these changes imply nuclear gene transcription, as much as 50% depend on the regulation of mRNA stability [16]. This indicates that the regulation occurs at the genomic level, but there is also a tight post-transcriptional control of gene expression essential for T lymphocyte activation, as in many other physiological processes. In fact, it is estimated that 30–90% of the mouse and human transcriptome is controlled by miRNAs [17,18]. In line with this, unbiased profiling using large qPCR panels, microarrays and deep miRNA sequencing identified specific miRNA patterns of expression that suggested an important role for miRNAs in cell lineage determination and effector functions in hematopoietic and lymphoid cells [19,20]. Genetic mouse models lacking either one or several of the key enzymes for mature miRNA biogenesis, namely Dicer, Drosha or DGCR8, exhibited reduced numbers and fitness of T lymphocytes, together with a skewed T cell response, with increased IFN-γ production and impaired proliferation rate [21,22,23]. Systematic approaches, using knockouts and conditional knockouts of individual miRNAs, together with miRNA gain-of-function and loss-of-function studies, allowed the further dissection of the roles of several individual miRNAs in the regulation of T cell proliferation, activation, and polarization towards the different subsets. Although very exhaustive and comprehensive reviews have deeply analyzed the role of miRNAs in T cell development, activation, differentiation and function [24,25,26,27,28,29], herein we summarize the best characterized T cell miRNA regulators that may be used as potential therapeutic agents.

### 2.1. miR-155

miR-155 is one of the most widely studied miRNAs with pleiotropic effects, in particular as a key regulator of T cell responses [30,31]. miR-155-deficient mice are characterized by a skewed CD4+ T differentiation towards Th2 [32,33], with increased secretion of the Th2 cytokines IL-4, IL-5 and IL-10 in vivo [34]. Moreover, cultured miR-155 knockout CD4+ T cells showed a decrease in *IFN*-γ expression in resting conditions, but remained unaltered if complemented with Th2 cytokines in vitro [34]. Importantly, miR-155 is upregulated upon T cell activation and has also been found to be required for optimal T-cell dependent germinal center response and antibody production [32,35]. Among the many targets identified, miR-155 inhibits *c-Maf* [34] and both *Socs1* and *Ship1* [36,37], with paramount roles in Th cell function. In addition, epistasis experiments showed that miR-155 is dominant over miR-146, since CD4+ T cells lacking both miRNAs reproduced the single miR-155-deficient mouse phenotype, with defective IFN-γ expression and antitumor immunity [38]. Additionally, miR-155 knockout mice reported a deficiency in Treg populations [39,40] due to *Socs1* 3′ untranslated region (UTR) negative regulation [41].

miR-155 is also required for the function of cytotoxic and memory T cells [32,34,42,43]. Its deficiency in CD8+ T cells results in reduced cytotoxicity [42] and decreased effector cytokine production [43]. Furthermore, miR-155 enhances the responsiveness of CD8+ T cells to the homeostatic cytokines, IL-7 and IL-15, as well as IL-2, which has a key role in tolerance and immunity [34]. miR-155 principal mRNA targets in cytotoxic T cells are similar to those described for Th cells, including Socs1 [43], Stat1 [44], Ship1 [37,45], Irf7 [46], and Ptpn2 [42], among others. Further experiments with chronic infection models showed that *Ship1* repression by miR-155 is sufficient to produce significant effects in resolving inflammation [47].

### 2.2. miR-146a

miR-146a is highly expressed in memory cells and induced upon T cell activation in human [29]. Moreover, miR-146a-deficient mice present hyperactivated lymphocytes and fail to resolve inflammation. miR-146 expression increases upon TCR engagement, leading to NF-κB repression, at least partially by targeting the 3′ UTRs of *Traf6* and *Irak1* mRNAs [28]. This leads to reduced levels of IFN-γ both in vivo and in vitro, acting as a powerful inhibitor of inflammation and autoimmunity. In fact, miR-146a also has a major role in Treg suppressor function, and conditional FOXP3+ miR-146a knockout mice are characterized by IFN-γ dependent and Th1 cell-mediated immune lesions. Possibly, these effects are dependent on enhanced expression of *Stat1* in miR-146 deficient cells, as they lack this key negative regulator [48,49,50].

### 2.3. miR-17~92

The miR-17~92a cluster is composed of six individual miRNAs (miR-17, miR-18a, miR-19a, miR-20a, miR-19b-1 and miR-92a-1), that can be grouped into four distinct miRNA families according to their sequences [51,52]. Although miR-17~92a is transcribed as a single transcript and highly induced upon CD4+ and CD8+ T cell activation [53,54], the individual miRNA cluster members are differentially processed post-transcriptionally [55]. Consistent with this association to T cell activation, miR-17~92a is overexpressed in peripheral CD4+ T cells of patients with several immune-related pathological conditions, such as multiple sclerosis [56], asthma [57], or breast cancer [58]. Additionally, it has a critical role in T cell polarization, acting as a positive regulator of Th1 differentiation and promoting anti-viral IFN-γ responses [59]. Indeed, Th2 polarizing environments induce downregulation of miR-17~92 [60], and its overexpression is sufficient to induce Th1 differentiation upon activation. This effect is mainly due to the function of miR-19b, which directly targets the negative regulator of *Pten* [53].

Interestingly, transgenic overexpression of the miR-17~92a cluster in mice leads to enhanced IFN-γ production, which is related to cytotoxic activity and lymphoproliferative disease [61,62,63]. Besides *Pten* [62], it has a direct interaction with the pro-apoptotic *Bim* [64], which leads to increased IL-4 levels [57].

Whereas the function of miR-17~92a in CD4+ and CD8+ T cells is well established, its role in Treg polarization is controversial since it has been linked to either inductor or inhibitory functions. Conditional miR17~92a downregulation in Treg cells leads to normal populations but deficient IL-10 production in autoimmune encephalitis mouse models [65]. Moreover, activation of CD4+ T cells in miR-17~92a-deficient mice induces the expression of the transcription factor Foxp3, which is a typical Treg marker. Conversely, miR-17 and miR-19b have been described as powerful suppressors of Treg differentiation [53,66].

### 2.4. miR-181

miR-181 is present in four different isoforms of mature miRNAs which are encoded by three independent clusters. One of these clusters, miR-181a1/b1, has a critical role in inducing thymocyte development [67,68,69] and CD4+ T cell stimulation [67,68]. Actually, miR-181 is essential for positive and negative selection in the thymus [70], since miR-181 deficient mice showed a 50% inhibition of negative selection and defects in positive selection, which were related to the increase of *Nrarp* [68]. Besides thymocyte development, miR-181a overexpression increases the sensitivity to peptide antigens in mature T cells [70], mainly by the inhibition of *Ifn*-*γ* [71], and different phosphatases that regulate TCR signals, such as *Shp2*, *Ptpn22*, *Dusp5* and *Dusp6* [70], while miR-181c-5p directly targets *IL-2* [72]. *Pten* inhibition by miR-181a1/b1 also showed important effects in natural killer (NK) T cell function, as shown by miR-181a1/b1 knockout mice that exhibited a deficient NKT population, which was rescued upon *Pten* silencing [67,69].

### 2.5. miR-21

miR-21 is involved in many biological processes and, therefore, is dysregulated in several pathologies, such as cardiovascular diseases, cancer and inflammatory diseases [73]. Patients with systemic lupus erythematosus present upregulated levels of miR-21, that promote aberrant T cell responses [74]. miR-21 directly inhibits IL-12 expression on dendritic cells, resulting in T-bet and IFN-γ-mediated induction of proliferation and survival of Th1 cells [75,76]. IL-4 release is also regulated by this miRNA, since CD4+ T cells from miR-21 deficient mice stimulated in vitro produce less IL-4 compared to controls [75]. Likewise, miR-21 controls IL-10 secretion by inhibiting the *Pdcd4* 3′ UTR 74. miR-21 is also enriched in murine Tregs, where it mediates a positive indirect regulation of Foxp3 expression [77,78].

Figure 1 summarizes some of the main miRNAs implicated in T cell function and differentiation which, in turn, may represent therapeutic targets for the treatment of pathological conditions where these subsets are dysregulated.

## 3. miRNAs in Immunotherapy

miRNAs can either boost or dampen immune responses, in physiological and pathological processes, where specific miRNAs have been associated with either the resolution or the progression of disease. Thus, the delivery of therapeutic miRNAs and/or antagomiRs has been extensively evaluated in several pre-clinical and clinical models, as reviewed in this section.

### 3.1. miRNA Function in Cancer

Abnormal miRNA expression has been widely associated with human cancer establishment and progression [6]. The outbreak of studies analyzing the expression of miRNAs as biomarkers for tumor progression and for disease prognosis during therapy has prompted researchers to investigate the effects of therapeutic miRNAs in human malignancies. Strategies to target either oncogenes or tumor suppressor genes [13,79], as a means to control tumor growth or to boost anti-tumoral immune responses in the tumor microenvironment, have been undertaken. Herein, we will focus on the therapeutic targeting of anti-tumoral T cell function.

In particular, given the importance of miRNAs for T cell regulation and polarization, it is not surprising that they are also related to lymphoproliferative diseases and other types of cancer, where the dynamic interaction of immune and tumor cells plays a critical role in controlling cancer progression. Immune checkpoints, such as PD-1 [80], its ligand PD-L1, and CTLA-4 [81], are key regulators of immunity. They are necessary to ensure efficient immune responses, while preserving tissue homeostasis. The blockade of immune checkpoint molecules has been extensively explored to re-activate anti-tumoral responses and revert T cell exhaustion. The use of monoclonal antibodies capable of neutralizing these immune checkpoints represents, nowadays, a powerful immunotherapy strategy for the treatment of several types of cancers. However, clinical trials have shown that not all patients benefit from these therapies, and emerging immunological strategies are being explored to restore T cell homing and function in the tumor microenvironment, and to recover immune function, including mononuclear phagocytes activity [82]. Importantly, miRNAs targeting immune checkpoints constitute an attractive therapeutic target in cancer treatment.

#### 3.1.1. PD-1 and PD-L1 Regulation by miRNAs

The PD-1 receptor is expressed on the surface of immune cells, including T lymphocytes, and can be triggered upon binding to its ligands PD-L1 and PDL-2, expressed on immune subsets and tumorigenic cells. PD-1 engagement inhibits TCR signaling, lymphocyte effector functions and clonal expansion [83,84,85]. Thus, the PD-1/PDL-1 axis is involved in T cell exhaustion, impairing T cell-mediated immunosurveillance in cancer and chronic infection. The recent discovery of PD-L1 transcriptional regulation brings into focus the use of miRNAs as a complementary treatment for traditional therapies [86,87]. Indeed, the expression of PD-L1 immune checkpoint, expressed by immune and tumorigenic cells, is tightly regulated at the post-transcriptional level through multiple miRNAs that bind to its 3′ UTR, resulting in translation repression. A number of miRNAs, such as miR-142-5p, miR-138-5p, miR-513, miR-570, miR-152, miR-200 and miR-34a, have been widely studied in the context of PD-L1 inhibition, expressed by immune and tumorigenic cells [88]. However, among all these individual miRNAs, the function of the miR-200 and miR-34a axes have been investigated in most detail and will be described in this section.

The miR-200 family includes five molecules (miR-200a, miR-200b, miR-429, miR-200c and miR-141) that participate in PD-L1-mediated epithelial-mesenchymal transition (EMT), a critical process for tumor metastasis [89]. In fact, there is a link in non-squamous cell lung cancer (NSCLC) between EMT and CD8+ tumor infiltrating lymphocyte immunosuppression [90]. Direct inhibition of tumoral PD-L1 by miR-200a results in an increase in tumor infiltrating T cells and a delay in metastasis. These effects could be reversed by overexpression of ZEB-1, an upstream suppressor of miR-200a [90]. These and other studies suggest that miR-200a may be a good biomarker for diagnosis of different types of cancer, such as lung, bladder, ovarian or breast cancer, as well as a potential adjuvant in immunotherapy vaccines in combination with anti-PD-1 or anti-PD-L1 antibodies [91,92,93,94,95].

miR-34a is also an important tumor suppressor. Remarkably, miR-34a targets many oncogenes related to cell proliferation, apoptosis and invasion, and several studies have shown that miR-34a therapy is a promising approach in cancer treatment [96,97]. miR-34a is known to be downregulated in in chronic lymphocytic leukemia [98], colorectal cancer [99], lung cancer [96,100], brain tumors [101], or prostate cancer [102], among others. Importantly, in a recent study, the miR-34 family was associated with the regulation of PD-L1 expression [103]. P-53 deficient cell lines showed decreased expression of both miR-34a and PD-L1 via miR-34a, and luciferase assays confirmed the transcriptional arrest of PD-L1 mediated by miR-34a [103]. The administration of MRX34, a liposomal miR-34a mimic, led to decreased levels of tumoral PD-L1 in NSCLC mice. Additional studies also demonstrated that miR-34a treatment in subcutaneous H460 xenografts was capable of inhibiting tumor proliferation and inducing apoptosis. miR-34a administration promoted the downregulation of its direct targets, *c-Met*, *Cdk4*, and *Bcl2*, correlating with diminished levels of protein expression [96]. Similarly, miR-34a inhibits PD-L1 in acute myeloid leukemia. In fact, transfection with miR-34a precursors resulted in the reduction of IFN-γ-induced PD-L1 surface expression in a dose-dependent manner in HL-60 cell lines [104].

Additionally, although the miRNA regulation on the PD-1/PD-La axis has been mainly described on PD-L1 expression, the 3′ UTR of *Pd-1* can also be targeted by miRNAs [105]. In particular, miR-28 was found to be capable of targeting PD-1 on T cells and therefore of modulating exhaustion and cytokine release [106].

#### 3.1.2. CTLA-4 Regulation by miRNAs

CTLA-4 is expressed on the surface of T lymphocytes during the initial stages of activation and upon TCR engagement and co-stimulation. Additionally, it is constitutively expressed on Treg cells [107]. Due to its homology to the CD28 receptor, CTLA-4 binds to the antigen-presenting cell (APC) receptors B7-1 and B7-2 [108,109], and this linkage provokes their internalization from the surface of APCs [110]. As a consequence, the essential co-stimulatory signal, which is normally provided by CD28, is lost, inhibiting T cell activation [111]. Early studies proved that the administration of antibodies against CTLA-4 results not only in tumor shrinkage but may also protect against tumor relapse [112,113,114]. Although pre-clinical results were very promising, checkpoint blockade did not succeed for all types of cancer and treatment failure was widely related to autoimmune side effects [115,116]. This pointed to the need for complementary approaches such as miRNA inhibitory therapies to benefit a higher percentage of patients [117].

Several miRNAs have been reported to directly modulate the expression of CTLA-4. For instance, miR-138 targets the 3′ UTRs of both *Pd-1* and *Ctla-4* mRNAs, inhibiting their expression. Additionally, the treatment with miR-138 in immunocompetent mice boosted anticancer immune responses, resulting in tumor shrinkage in murine models of glioma [117]. Moreover, aberrant expression of miR-138 was related to fulvestrant and tamoxifen resistance in mouse models of breast cancer [116]. miR-487a-3p is another example of a direct regulator of CTLA-4 translation. Analysis of public databases underscored decreased levels of miR-487a-3p in prostate cancer patients and type I diabetic patients [118], that were further confirmed by in situ hybridization and qRT-PCR [119]. Overexpression of this miRNA led to defects in tumor cell proliferation, cell cycle, migration, and invasion, promoting a significant reduction of tumor size in xenograft mice models [120,121]. Additionally, miR-9 acts as an inhibitor of Treg cell activation, through direct inhibition of *Ctla-4*, *Foxp3* and *Garp*, as demonstrated with site-directed mutagenesis and luciferase experiments [121]. Nevertheless, the role of miR-487a-3p and miR-9 in cancer remains controversial as they have also been reported as pro-oncogenic miRNAs in hepatocellular carcinoma [119] and different types of cancer [122].

### 3.2. miRNA Function in Immune-Related Diseases

miRNAs play a pivotal role in immunity and inflammation. The association of miRNAs with disease, and their predictive value for prognosis and relapse after treatment, has been established in several immune-related pathologies. Some examples are asthma [57], systemic lupus erythematosus [123] and lupus nephritis [8], rheumatoid arthritis [124,125], autoimmune type 1 diabetes mellitus [126,127], or multiple sclerosis [128], among others.

Hence, several studies have explored the therapeutic potential of miRNAs and antagomiRs to restore immune homeostasis in pre-clinical models of several immune diseases and inflammation. Patients with primary Sjögren’s syndrome, an autoimmune disorder accompanied by systemic inflammation and lymphocytic infiltration (mainly T cells) of the exocrine glands, have increased levels of miR-744-5p at the ocular surface. Administration of antagomiR-774-5p reduced the levels of the pro-inflammatory IFN-dependent chemokines CCL5 and CXCL10 via *Pellino3* downmodulation [129]. Similarly, miR-130b-3p delivered by mesenchymal stem cells-derived exosomes was found to limit LPS-induced acute lung injury in murine models by targeting *Tgfbr1* [130].

miRNAs have been shown to be master regulators of T cell responses, as reviewed in Section 2. A number of pre-clinical studies have focused on siRNA-directed T cell targeting to control immune pathologies involving T cell dysregulation. In 2008, a seminal study paved the way for the therapeutic use of siRNAs to modulate T cell immune responses. Targeted stabilized NPs (tsNPs) containing Cyclin D1 siRNA reversed experimentally induced colitis in mice by suppressing leukocyte proliferation and Th1 cytokine expression [131].

As above-mentioned, Tregs play a pivotal role in maintaining homeostasis and self-tolerance by suppressing the immune response. The impairment of their function, which is tightly controlled by miRNAs, leads to immune-related diseases and cancer [132]. miR-27 has been recently shown to regulate Treg-mediated immune tolerance [133]. A number of articles described the key role of miRNAs in regulating Th17/Treg balance in experimental autoimmune uveitis, as reviewed in [134], highlighting miR-223-3p, miR-155 and miR-146a as potential therapeutic targets. Moreover, Tregs release miRNA-containing exosomes, bearing let-7d, that contributed to the suppression of pathogenic Th1 cells, preventing systemic disease [135]. Another study identified miR-10a and miR-182 as critical modulators of Th1 subsets, after *Leishmania major* infection, or Th2-associated Treg cell function, following *Schistosoma mansoni* infection [136]. Similarly, miR-155 was required for effective type-2 immunity, as highlighted by deficient mice studies, in house dust mite-allergic or helminth-infected animals [137]. Recent advances also highlight the potential of miRNAs for the treatment of asthma [138]. Importantly, treatment with cell-penetrating peptide (CCP)-miR-146a nano-complexes had a potent anti-inflammatory function, reducing allergic inflammation in house dust mite models and *Rhinovirus* infection [139]. Similarly, miR-126 was recently described to be involved in the development of allergic rhinitis, modulating the ratio of Tregs and effector Th1/Th2 cells. Treatment with either miR-126 mimics or antagomiRs was capable of regulating T cell subsets polarization and cytokine release related to the pathogenesis [140]. In line with this, a recent report showed that NK-cell-derived EVs were enriched in miRNAs related to Th1 polarization. miR-10b, miR-92a and miR-155 induced Th1 differentiation in CD4+ T cells, but also had an impact on monocytes and DCs by activating their polarization, presentation and co-stimulatory capacities. Furthermore, tailored gold nanoparticles (NPs) bearing these miRNAs were capable of promoting Th1-like responses in vivo and they activate T cell lymphocytes [141].

miR-210 genetic ablation, and antagomiR-210 intradermal injection, were capable of blocking T cell inflammatory skewing and the development of psoriasis-like inflammation in mouse models [142,143]. Two independent studies demonstrated the role of miRNA delivery in suppressing inflammatory bowel disease, miR-219a-5p by inhibiting Th1/Th17 responses [144], and miR-106 inhibition by inducing Treg suppressive function and promoting IL-10 release [145]. An independent study identified miR-467b as a potential target to alleviate experimental autoimmune encephalomyelitis, by inhibiting the differentiation and function of Th17 cells via *eIF4E* targeting [146].

miRNAs have also been explored as potential targets to boost immune responses against several infectious diseases. siRNAs against CCR5 to block viral entry, together with a mixture of antiviral genes, were selectively delivered to T cells, using a CD7-specific single-chain antibody conjugated to oligo-9-arginine peptide. This formulation was capable of suppressing HIV-1 viremia in humanized infected mice [147]. Extracellular vesicle-transfer of miR-139-5p, has been shown to promote activation of CD4+ HIV-infected cells upon targeting of *Foxo1* and the PD-1/PD-L1 promoters *Fos* and *Jun*, being a potential therapeutic target to treat HIV patients and block the reactivation of virus latently infected T cells [148]. Additionally, miR-155 was found to play an important role in T cell immunity against *Toxoplasma gondii* [149] and *Trypanosoma cruzi* infection [150].

Besides T cell modulation, miRNAs are key players in the development and function of other immune cells, including B lymphocytes [151,152,153] and macrophages [154,155,156], among others. Macrophages are another important immune population whose function is dysregulated in several pathological conditions. During tumor progression, the protective M1 phenotype shifts towards the pro-tumorigenic M2-phenotype [157]. Targeting macrophages to skew M1/M2 polarization, by delivery of immunoregulatory miRNAs/antagomiRs, is emerging as a novel approach for the treatment of several diseases that involve dysregulated macrophage function [158]. Accumulating evidence indicates that miRNAs are molecular switches in macrophage activation and polarization [159], e.g., miR-155, miR-181a, and miR-451 [159]. Pre-clinical studies that explore the specific delivery of these macrophage-polarizing miRNAs have been carried out in a variety of disease models, such as abdominal aortic aneurysms [160], choroidal neovascularization [161], rheumatoid arthritis [162], or cancer progression [163].

A better understanding of the immunomodulatory functions of individual miRNAs may be crucial to design effective therapies to restore dysregulated immune cell function in disease.

### 3.3. miRNAs in Clinical Trials

The number of clinical trials involving the use of miRNAs has exponentially increased in the last few years, with 1.188 studies registered to date (https://clinicaltrials.gov accessed on 16 December 2022). However, most of these studies are observational and involve analysis of body fluids with a putative diagnostic and/or prognostic value to monitor disease progression, while 565 studies are listed as interventional. Several clinical studies include the direct administration of miRNAs, such as miR-16 (NCT02369198), miR-29 (NCT03601052), and miR-34 (NCT01829971). Conversely, anti-miR-21 (NCT03373786), anti-miR-92a (NCT03603431), and anti-miR-122 (NCT01200420) are examples of clinical trials focused on the potential of antagomiRs for treatment.

While some pre-clinical in vivo effects are very promising, results in clinical trials to date remain inconclusive but open encouraging perspectives. miR-34a mimic (MRX34) administration using liposome vehicles has been tested in two phase 1 clinical trials with hepatocellular and NSCLC patients [97]. Although the first attempts raised safety concerns due to severe immune-related adverse events [97,164], pharmacodynamic analysis in MRX34-treated patients showed downregulation of miR-34a-relevant target genes in white blood cells and increased levels of miR-34a in tumor tissue, providing proof-of-concept for miRNA-based cancer therapy. Furthermore, one patient with hepatocellular carcinoma achieved a prolonged confirmed pathologic response that lasted for four years, while four patients demonstrated stable disease for at least sixteen weeks [103]. Pre-administration of dexamethasone increased the tolerance in a subset of forty-seven patients bearing solid tumors refractory to standard treatments; however, whether the effects are due to miRNA mediated PD-L1 silencing or immune-mediated antitumor activity remains unknown. For instance, the sequence of miR-34a, enriched in GU nucleotides, and the unknown chemical formulation of MRX34 cannot be ruled out as responsible for Toll-like receptors stimulation and require further investigation [164].

Remarkably, intradermal treatment with Remlarsen, a miR-29 mimic, in forty-seven healthy subjects repressed collagen expression and the development of fibroplasia in incisional skin wounds [165]. Importantly, in this study, only seven individuals experienced reactions of short duration which were easily solved without medical intervention.

TargoMirs, minicells loaded with miR-16 mimics, were also used in another clinical trial, as a means to suppress tumor growth, dampened in malignant pleural mesothelioma murine models [166]. miR-16-TargoMir was administered to twenty-six patients with malignant pleural mesothelioma in a phase 1 trial. This trial showed a favorable safety profile; however, the miR-16 biodistribution was not analyzed in this study [167]. Notably, ABX464, a long non-coding RNA which, through splicing, can overexpress miR-124, exhibits antiviral effects. Treatment of HIV-infected patients [168] showed some reduction in viral load [169], although further studies are required to confirm treatment efficiency. In an additional trial in ulcerative colitis patients, good results were reported at all doses with very mild adverse effects and a phase 3 clinical study is currently ongoing [170].

It is also worth mentioning the high cure rates in chronic hepatitis C patients, after subcutaneous injection of RG-101 (anti-miR-122) in combination with the administration of the viral protein inhibitor GSK2878175 [171]. Treatment was well tolerated and all patients showed a substantial viral load reduction within the first month, and a sustained antiviral response in several subjects [172]. Nonetheless, RG-101 development was arrested owing to adverse effects observed in a different clinical trial [172]. Additionally, Cobomarsen (anti-miR-155) was used in different clinical trials in patients with cutaneous T lymphoma but with still inconclusive results.

Overall, substantial advances in miRNA therapies have led to a number of clinical trials, summarized in Table 1, with promising results. However, most clinical trials had to cope with adverse effects related to their administration. Noteworthily, most clinical studies to date appear to use chemically modified miRNAs without specific delivery systems, except MRX34, which was delivered in liposomes. Deficiencies in tolerability, immunogenicity, specificity, pharmacokinetics, and delivery efficiency of the miRNAs were reported in several studies. To solve these difficulties, enormous advances have been achieved in the last few years, mainly through the combination of miRNAs with traditional therapies and through the optimization of new types of delivery systems, as reviewed in the following sections. However, multidisciplinary improvements are essential to implement miRNA therapies as a consolidated treatment [173].

## 4. Non-Nano Based Strategies for miRNA Delivery

The pivotal modulatory function of miRNAs in homeostasis and disease has prompted researchers to implement targeted delivery strategies to promote efficient and specific gene regulation within specific tissues and cell types, while avoiding miRNA degradation and off-target effects.

### 4.1. Engineered EVs

Extracellular vesicles are double lipidic bilayers naturally present in biofluids such as blood, cerebrospinal fluids and urine. They are categorized as exosomes, apoptotic bodies or microvesicles depending on their size, biogenesis, and marker expression [180]. Regardless of their size, all EVs carry different types of cargoes such as proteins, lipids and genetic material, including miRNAs. EVs are enriched in specific small RNAs compared to producing cells [181,182] and it has been demonstrated that small RNAs are more likely to be actively exported into EVs than mRNAs [183,184].

Owing to their bioactive content, endogenous EVs have been widely studied as mediators of intercellular communication and, in particular, they have been related to several pathological processes, such as cancer [185]. EVs act as a shield, keeping the miRNAs or antagomiRs intact when transferred to recipient cells [186], making them potentially useful for therapy. Importantly, EVs present important benefits, such as circumventing the host immune surveillance due to their biological origin, their capacity to cross biological barriers, and the high efficiency of delivery to bystander or distant cells [187]. In addition, they are relatively easy to produce on a large scale and their content can be modified [188], e.g., to target specific tissues and/or cell types, which make them very versatile. Despite their important advantages, some of their major limitations for therapy are their rapid clearance after administration and their variable and non-controlled content [189]. To solve this, several engineering strategies have been developed, e.g., EV decoration with albumin, that increases their circulation lifetime and tissue residency [190].

The EVs content can be enriched in one specific molecule through pre-loading (parental manipulation of the cell) or post-loading (EV modification after isolation) techniques. The pre-loading approach consists of overexpressing the molecule of interest in the cell of origin, either by transfection, co-incubation or gene modification, followed by EVs isolation. Post-loading enrichment, in contrast, relies on the incorporation of the molecules in already isolated EVs. For this, it is necessary to induce the formation of transient pores to increase the membrane permeability. Electroporation, sonication, extrusion, co-incubation, freeze–thaw cycles, saponin treatment and click chemistry are examples of post-loading approaches [191,192]. Interestingly, a recent and innovative method, combining nanotechnology and EV engineering without disrupting their membrane, has been reported. This method provides an efficient methodology to achieve a high load of catalytically active ultrathin palladium nanosheets inside exosomes for targeted bio-orthogonal catalysis, without damaging membrane integrity, based on a mild reduction process using gas-phase CO [193].

Recently, a number of studies reported the therapeutic potential of engineered EVs. Transfection of tumoral-derived EVs with exogenous let-7i, miR-142 and miR-155 mimics, before injection in tumor-bearing mice, leads to increased dendritic cell maturation, T cell activation and tumor reduction [194]. In addition, slowed tumor growth related to PD-L1 reduction was reported after intra-tumoral administration of miR-424-5p-enriched EVs [195]. Another study reported the efficient use of tumor-derived exosomes (TEX) enriched in miR-124-3p mimics using saponin-based approaches [196], for colorectal cancer treatment in pre-clinical models. After subcutaneous injection of miR-124-3p-TEX, the tumor size was reduced, associated with a higher survival rate through the modulation of Tregs, infiltrating T cells and splenocytes. Additionally, another study used modified M1 macrophage-derived exosomes, coated with IL-4 receptor and enriched with NF-κB p50 siRNAs and miR-511-3p [197]. This EV formulation reported a rise in M1 cytokines and immune-stimulatory cells compared to untargeted and control peptide-labeled exosomes. Furthermore, tumor growth was inhibited upon EV treatment, presumably by tumor-associated macrophage reprogramming into M1-like macrophages and increased anti-tumor immunity.

### 4.2. Cationic Polymers

Cationic polymers have also been extensively used for nucleic acid delivery. Once positively provided, they can be conjugated to the negatively charged nucleic acids, forming linear or branched/dendritic polyelectrolyte complexes. In addition, cationic polymers are biocompatible, biodegradable, flexible, come from renewable resources and possess low immunogenicity, making them good candidates for gene delivery. However, poor gene-transfer efficiency, due to high enzymatic degradation rates and endolysosomal escape, have limited their clinical application [198]. Some examples of naturally-derived cationic polymers are chitosan, dextran, gelatin, cellulose, and cyclodextrin polymers [199]. Nevertheless, the capacity of cationic polymers as miRNA carriers has been scarcely reported, and only a few studies have explored their carrier potential in vivo. In particular, chitosan/miR-124 polyplex particles were transfected in microglia cells ex vivo, resulting in an effective reduction of reactive oxygen species and TNF-α/MHC-II molecules. Importantly, in vivo peritoneum administration of these particles was effective as they arrived to the spinal cord injury three days post-injection with a significant decrease in neuronal inflammation [200]. In another study, multiple β-cyclodextrin-attached quantum-dot based particles were loaded with 5-fluorouracil and miR-34a mimics. These carriers were effectively delivered to colorectal cancerous cells both in vitro and in vivo. Moreover, they reduced proliferation and migration rates, resulting in a decrease in tumor size [201]. In conclusion, despite the advantages of cationic polymers, their low transfection efficiency highlights the need for optimization of these carriers to be used as miRNA delivery agents. Synthetic polymers, a good alternative to cationic polymers, will be further discussed in Section 5.2.

### 4.3. Viral-Based Delivery Systems

Viruses have been widely used as delivery vectors to insert genetic material (DNA/RNA) into host cells. This delivery strategy consists of using engineered viruses, such as adenoviruses, adeno-associated viruses, lentiviruses, or retroviruses, in which virulence-related genes are removed, while the genes of interest are inserted, e.g., miRNA cassettes [202,203]. Viral-based systems constitute an efficient strategy to deliver miRNAs; however, their systemic toxicity and immunogenicity limits their clinical use [203].

#### 4.3.1. Adenovirus and Adeno-Associated Virus

Adenoviral vectors have attracted attention as delivery tools owing to their capacity for transducing a variety of cells, both quiescent and dividing, without integrating their viral cargo into the host genome [204]. However, one of the principal drawbacks of their use is their potent activation of immune responses and cell toxicity [205,206]. The use of gutless adenoviruses has helped to reduce immune-mediated toxicity [207]. Additionally, adenoviral treatment usually requires repeated administration, which limits their long-term therapeutic use, but it can be suitable for short-term use, since its repression of gene expression has been shown to last for up to five weeks [208]. In 2002, the first study using adenoviruses to deliver interfering RNAs to cells, both in vitro and in vivo, was carried out [209]. Adenoviral vectors efficiently reduced the expression of target genes in the liver and the brain, indicating that they could be useful to treat hepatic and nervous system diseases. Moreover, they are versatile and efficient in the co-delivery of miRNAs and proteins in various in vivo models, e.g., viral infection [210,211], or vascular-related diseases [212,213], among others. Importantly, the growing interest for gene therapies has led to the commercialization of several adenoviral-based products, including oncolytic viruses, that predominantly kill tumor cells, and COVID-19 vaccines, e.g., Astra Zeneca, with satisfactory results for review [214].

To increase the specificity and minimize off-target effects and toxicity, recombinant adenoviruses with deficiencies in replication, adeno-associated non-enveloped viruses, or conditionally replicating adenoviruses have been studied and included in clinical trials as potential treatments for cancer or vaccines [211,214]. However, immunogenicity remains one of the main shortcomings for these types of viruses [215], due to a strong activation of both innate and adaptive immune responses in the host [216]. Several strategies are being explored to overcome this limitation, such as viral capsid modifications, but it is worth mentioning that adenovirus-mediated immune boosting may also be beneficial for cancer therapies or vaccines to fight against infectious diseases, as highlighted by an increased effectiveness of SARS-CoV-2 vaccines [217].

#### 4.3.2. Retrovirus

Retroviruses have been also analyzed as vectors for miRNA delivery. In this case, the viral RNA genome integrates randomly into the host genome, which is advantageous for the stability of gene expression. However, transgenes may be transcriptionally silenced over time [218] and RNA integration may compromise safety. In this sense, retrovirus insertion in unwanted genome sites is an important concern for the safety of their use as therapeutics, and have been linked to the development of leukemia in clinical trials [219,220]. Although retroviruses induce discrete immune responses in the host, compared to adenoviruses, the main limitations for their use as delivery vectors rely on their safety concerns, their low inserting capacities and vector titers, together with their restricted tropism and selective incorporation in dividing cells [203]. Despite the important concerns for the safety of the use of retroviral systems for human therapy, clinical studies showed that ex vivo transduction of CD4+ T cells, followed by re-infusion of transduced cells, was safe in phase 1 clinical trials [221]. However, phase 2 studies failed to deliver anti-HIV viral ribozymes efficiently [222], and although the use of retroviral vectors for miRNA delivery has been explored, the important drawbacks for their use have shifted the interest towards other strategies.

#### 4.3.3. Lentivirus

Lentiviruses, as retroviruses, integrate into the genome, but are able to transduce both dividing and non-dividing cells, and importantly exhibit a better safety profile than retroviruses, with a lower risk of insertional mutagenesis [223,224]. However, they do exhibit some limitations, such as modest insertional capacity and low vector titers and risk of mutagenesis upon insertion [203]. Several phase 1 clinical trials have documented the safety of lentiviral-based therapies, and the stability of vector expression [225,226], with limited therapeutic effects. Recently, lentiviruses have emerged as a very promising therapeutic tool for haemopoietic stem cell gene therapy, such as for the treatment of metachromatic leukodystrophy [227] and Wiskott–Aldrich syndrome [228,229]. Several pre-clinical models have shown efficient delivery of miRNAs using lentiviruses and therapeutic effects in various types of cancer [230,231,232,233] or arthritis [234].

## 5. Nano-Based Strategies for miRNA Delivery

Nanotechnology offers exciting perspectives for the controlled release of miRNAs, allowing most of the hurdles for their therapeutic use in clinics to be overcome, including non-specific or inefficient uptake by target cells, undesired off-target or on-target effects, short lifespan in systemic circulation, limited stability, or cytotoxicity [235]. Moreover, NPs reach tumor tissues more efficiently than healthy tissues, benefitting the enhanced permeability and retention effect [236].

### 5.1. Lipid-Based Polymers

Liposomes are colloidal particles that have an aqueous core enclosed by one or more phospholipid bilayers or lamellae. Commonly, they are classified on the basis of their size (small, large and giant vesicles), number of bilayers (uni-, oligo- and multi-lamellar) and phospholipid charge (neutral, anionic or cationic) [237,238]. Liposomes are frequently formed of phosphatidylcholine complemented with fatty acyl chains. Additionally, it is usual to introduce cholesterol to increase rigidity and reduce serum-induced membrane instability [239,240]. Liposomes are one of the most used transfection reagents in vitro, due to their biodegradability, biocompatibility and their high resemblance to the cell membrane [241,242]. Nevertheless, some studies have reported high toxicity rates in liposomes, alongside non-specific uptake and the triggering of unwanted immune responses [243,244]. It is worth mentioning that some of these drawbacks, such as low specificity, can be easily overcome by surface modification. For instance, PEGylation of liposomes has been shown to increase half-life from minutes to hours in the bloodstream [245]. Additionally, pre-miR-133b delivery in cationic lipoplexes (lipids and nucleic acids complexes) was shown to be more efficient than control standard transfection agents (siPORT NeoFX) for lung delivery in mice models [246]. Importantly, liposomes can be designed to release their contents in acidic environments, as endosomes and lysosomes, using pH-triggered approaches [247].

Several works which relate efficient liposome-based miRNA delivery with tumor inhibition have been recently published. For instance, intraperitoneal administration of 1,2-dioleoyl-sn-glycero-3-phosphatidylcholine (DOPC) nanoliposomes enriched in miR-192 leads to reduced angiogenesis and tumor regression compared to control and anti-VEGF antibody treatments [248]. These effects were then related to miR-192 direct inhibition of the angiogenic factors *Egr1* and *Hoxb9*. Notably, rescued Dicer expression and decreased tumor growth and metastasis were reported in vivo after DOPC-nanoliposome delivery of anti-miR-630 in combination with anti-VEGF antibody treatment [249]. DOPC nanoliposomes are already being tested in clinical trials, although these data remain unpublished. Similar effects were observed after lipidic-based delivery of the tumor suppressors let-7 and miR-34a administration in NSCLC mouse models [250].

It is also worth mentioning the extensive use of lipid NPs (LNP) after the success of Moderna and Pfizer’s delivery of mRNA-LNP SARS-CoV2 vaccines [251]. Although these vaccines reported some mild adverse effects, they have proven highly protective against SARS-CoV-2-related diseases. Their structure is very similar to liposomes, but not necessarily formed by a continuous lipid bilayer and slightly bigger in size [252]. Besides vaccines, they have also been used for cancer treatment with good results. For example, pre-miR-107 LNP administration in head and neck squamous cell carcinoma was more efficient than free pre-miR-107 in reducing tumor volume [253]. In another study, *Pik3r2* and *Pten*, targets of miR-126-3p and miR-221-3p, respectively, were efficiently inhibited after antagomiR administration in mice bearing lung cancer, reducing tumor burden and metastasis [254]. Similar anti-tumoral effects were obtained in hepatocellular carcinoma mouse models, after administration of miR-30a-5p [255], anti-miR-17 [256], or miR-122 [257]. Additionally, the treatment with LNP harboring miR-634 in pancreas xenograft mice [258], or miR-186 carried by lipopolyplex NPs dressed with GD2 in neuroblastoma pre-clinical models [259], showed a marked reduction in tumor size. In line with this, miR-26a, miR-130a and anti-miR-155 mimics in LNPs coated with anti-CD38 antibodies were efficiently delivered to leukemic cells and increased apoptosis in vitro. After administration in chronic lymphocytic leukemia mouse models, the most efficient treatment was the miR-26a, related to a greater downregulation of their targets [260].

### 5.2. Synthetic Polymers: PEIs, PAMAMs and PLGAs

Synthetic polymers ensure the stability of nucleic acids by facilitating the cellular uptake and RNA integrity in different fluids. Furthermore, they are highly tunable, allowing increased biodegradability and biocompatibility [261,262,263]. There has been a huge increase in the number of synthetic polymers for immunotherapy in recent years, and new combinations to deliver multi-therapeutic agents, variations of their chemical synthesis and functional modifications are constantly being investigated [264]. Below, we describe some of the most important and widely used nowadays.

#### 5.2.1. Polyethylenimine (PEI)

PEIs are polycationic polymers that, due to their amino density, have high DNA binding efficiencies and good transfection capacity. In fact, this was one of the first types of nanocarriers commercialized. Additionally, PEIs can be modified by adding mannose, galactose, transferrin or antibodies to obtain tissue-specific deliveries [265,266,267,268].

Some studies have reported efficient delivery of miRNAs using PEI nanocarriers. For instance, miR-24 was efficiently delivered associated with PEI NPs in a mouse model of acute myocardial infarction, promoting the inhibition of its target *Bim* and an improvement in ventricular remodeling and cardiac function [269]. Additionally, miR-145 has been efficiently delivered into mice bearing colorectal carcinoma, and in vitro to breast cancer cells, showing anti-tumoral effects and biocompatibility [270]. Individual delivery of miR-33 and miR-145 in low molecular PEI NPs showed good delivery efficiencies in a xenograft colon carcinoma mouse model. The increase of miRNAs in a tumor environment was coupled with a reduction of proliferation and increment in apoptosis [270]. miR-708-5p PEI NPs also showed good results in an NSCLC mouse model, not only as a therapy but also as a preventive approach [271].

Nevertheless, the synthetic composition of PEI can affect its buffering capacities and other properties, making them suboptimal for gene delivery. In particular, non-biodegradability and high positive charge density are the main threats to cell viability. The molecular weight and structure of PEIs affect their resistance to degradation and, therefore, their toxicity. Low molecular weight PEIs are less toxic than their high molecular weight counterparts, though low molecular weight PEIs show poor transfection efficiencies [272,273]. Besides polymer size, changes in the sequence are essential to overcome delivery limitations. Therefore, the linkage with other polymers or chemical treatments that change the buffering nature of PEIs has been explored. For example, reaction with acetic anhydride permitted lower acetylation rates that resulted in a marked efficiency improvement [274]. Similarly, alanine addition, dodecylation and hexadecylation improved gene delivery compared to standard PEI [275]. In line with this, poly-arginine PEGylated PEI NPs loaded with miR-145 were used in a prostate cancer model, showing enhanced uptake, tumor shrinkage and prolonged lifespan in vivo [276]. Association of miR-21 with poly-l-lysine-PEI NPs also reported good results in studies with breast cancer cell lines [277]. In another study, miR-603 was associated with PEI and then encapsulated in liposomes decorated with PEG and integrin receptors [278]. miRNA-PEI-liposome delivery, both in vitro and in vivo, to glioblastoma cells showed increased specificity compared to controls. The association of PEI with polyacrylic acid was effective for miR-22 transport to mouse models of vascular injury [279].

#### 5.2.2. Polyamidoamine Dendrimers: PAMAM

Polyamidoamine dendrimers are repeatedly branched macromolecules composed of a central core, interior branches, and an exterior surface with functional surface groups [262]. The synthesis process can be repeated for several times (‘generations’) to obtain complex structures. Here lies the benefit of dendrimers: the larger the structure is, the more coupling sites for active molecules. PAMAMs are the most studied type of dendrimers due to their biodegradability, their spheroidal structure and the large number of secondary and tertiary amines on the polymer. Again, cytotoxicity and low transfection efficiencies are the major hurdle for these polymers. Studies with partially degraded PAMAMs showed better efficiency results than non-modified PAMAM. This suggests that partially degraded dendrimers are more flexible, therefore allowing better linkage to the cargoes, complemented with a higher stability in solution [263].

A number of publications report PAMAM NPs as good miRNAs delivery carriers. A compendium of the most relevant works of recent years was gathered by Ban et al. [275]. For example, anti-tumoral effects were shown after injection with miR-22 and miR-150 PAMAM NPs in leukemia progression [280,281]. Once the cytotoxic effects and transfection deficiencies have been solved, PAMAM NPs could be potential miRNAs carriers for upcoming clinical trials.

#### 5.2.3. Poly Lactic-co-glycolic Acid (PLGA)

PLGAs are copolymers formed by a glycolic acid and a lactic acid linked through an ester bond. They are widely used for drug and nucleotide delivery because of their biodegradable and biocompatible properties [282]. Once inside the cell, PLGA NPs are hydrolyzed, generating glycolic acid and lactic acid which enter in the Krebs cycle and are degraded naturally [283]. In fact, varying the amount of each compound can change the degradation rate from months to years, making them strongly useful for clinical use [235]. For instance, low molecular PLGAs enriched in glycolic acid are hydrophilic and, subsequently, prone to degradation. Conversely, high molecular PLGAs are more hydrophobic and degrade more slowly than small ones. Although some studies reported the use of high molecular PLGA NPs [284,285], their hydrophobic nature coupled with their negatively conferred surface hinders the encapsulation of nucleic acids. Therefore, combination with positively charged compounds has been investigated to overcome these limitations. The linkage to CS, a cationic polymer, has been used with good results. miR-34a-CS-PLGA nanoplexes (drug nanoparticle complexes with oppositely charged polyelectrolytes) were systemically administrated in human multiple myeloma xenografts NOD-SCID mice, leading to increased lifespan and reduced tumor volume for 18 days. These results were confirmed with high transfection efficiency and low organ toxicity [286]. PEGylated coating of miR-122 PLGA NPs was shown to increase the permeability and retention in biological fluids lasting until 28 days. Additionally, PEGylation of PLGA NPs with miR-21 and gemcitabine was shown to be more efficient in vitro compared to control miRNA mimics [287,288]. In another study, PEI-PLGA-HA NPs loaded with antagomirs of miR-542-3p and the chemotherapeutic drug doxorubicin were incubated in triple negative breast cancer (TNBC) cells. Again, administration led to high encapsulation rates, reduced degradation in serum and apoptosis in targeted cells [289]. Another option is the implementation of peptide nucleic acids (PNA) as substitutes of antagomiRs. These PNAs were firstly described as short sequences complementary to nucleic acids in which the sugar phosphate backbone was replaced by a peptide [290]. In the miRNA delivery context, encouraging results have been obtained after conferring a positive charge on the antago-miR and stabilizing the interaction with PLGA [291]. For instance, PNA/phosphonothioate-PLGA NPs were used to target specifically both miR-155 and miR-21 in lymphoma cell lines. The delivery of antagomiRs was efficient, downmodulating both miRNAs ex vivo, and led to a reduction in viability. Moreover, the same approach was efficient for miR-141-3p delivery in ischemic stroke mouse models [292].

### 5.3. Natural Polymers: Hyaluronic Acid, Chitosan and BSA

Chitosan is a biodegradable and biocompatible polymer that has been intensively studied as a nanocarrier, owing to its easy preparation and its capacity to cross mucosal barriers. Due to its positive charge, chitosan easily forms complexes with anionic miRNAs under mildly acidic conditions, protecting miRNAs from degradation [293].

At present, treatment of TNBC mainly depends on chemotherapy with mild toxic side effects, but the effect is limited and highly prone to generate drug resistance. Due to the poor cell permeability and significant in vivo degradation rate of miRNAs/antagomiRs, which limit their clinical application, a core–shell supramolecular nanovector of “chitosome” was developed. The constructed chitosomes were capable of co-delivering hydrophilic anti-miR-21 and hydrophobic docetaxel (DTX), with an entrapment efficiency of more than 80%, spherical morphology, and average particle size of 90 nm. Anti-miR-21 encapsulated within chitosomes showed significantly increased cellular transfection and stability against degradation by nucleases in serum. Compared with DTX or anti-miR-21 formulations used alone, the delivery of the two drugs in chitosomes showed improved chemosensitivity of TNBC cells to DTX treatment through their synergistic effects. Taken together, chitosome could be a promising candidate for simultaneous delivery of insoluble chemotherapeutic drugs and gene agents for TNBC therapy [294].

Interestingly, another study developed a chitosan-based, self-assembled nanosystem that co-delivered miR-34a and doxorubicin with hyaluronic acid modifications to reverse the resistance of breast cancer cells to doxorubicin [295]. This system efficiently protected from nuclease degradation, and transported miR-34a and doxorubicin into drug-resistant cells. In addition, NPs were capable of inhibiting proliferation and promoting apoptosis by regulating the protein expression of Bcl-2 and PARP. Moreover, invasion, metastasis and adhesion were inhibited, by regulating E-cadherin, N-cadherin, MMP2, CD44, and Snail molecules [296].

Other natural polymers, such as bovine serum albumin (BSA) NPs, have also been explored as delivery nanocarriers as they are non-toxic, non-immunogenic, biocompatible, and can easily bind drugs, especially proteins, with high affinity [297]. However, very few studies have explored their use as RNA-carriers [298].

### 5.4. Inorganic NPs

Several inorganic materials have been used as nanotherapeutic agents, owing to their biocompatibility and their versatility to control loading, size or morphology for miRNA targeted release [248]. These include gold, calcium phosphate, silica, iron oxide and magnetic NPs, as extensively revised by Sekhon et al. [299,300].

Magnetic NPs have initially attracted interest for their use as contrast agents for magnetic resonance imaging, but their combination with cationic compounds allows efficient miRNA encapsulation, showing enhanced transfection efficiencies and combining the beneficial effects of miRNA delivery and static magnetic field or hyperthermia for therapy [301,302]. This technology has enabled the designing of promising therapeutic approaches in pre-clinical models of cancer [302], to promote bone regeneration and angiogenesis [303], wound healing [304], or immune modulation [305].

Calcium phosphate NPs have also been investigated as miRNA nanocarriers, since they are easily synthesized, cheap, biocompatible, and non-toxic [235]. However, miRNAs are not easily encapsulated in these NPs because of their low spatial charge density and, therefore, may not be the best approach for miRNA delivery.

Silica and mesoporous silica NPs (MSPs) have received great attention due to their high biocompatibility and stability. MSPs have been shown to be efficient carriers of miRNAs and are capable of co-delivering other therapeutics, such as anti-tumoral drugs [306] or surface molecules to enhance target delivery [307]. MSP-delivered treatment exhibited therapeutic effects in pre-clinical studies of cancer [308,309], or cardiovascular diseases [269]. Additionally, MSPs could modulate osteoimmune responses per se, although the mechanisms underlying these effects are not fully understood [295].

Gold NPs (Au-NPs) offer several advantages for therapy, including negligible toxicity, ease of functionalization with nucleic acids, and tunable shape and size [235]. It is also worth mentioning that gold NPs have been approved by the FDA and have shown great promise in a variety of medical applications [310]. Au-NPs were used to restore the tumor suppressor miR-145 levels in prostate and breast cancer cells [311]. Interestingly, gold–iron oxide NPs loaded with therapeutic miRNAs for glioblastoma have been administered intranasally in combined pre-clinical treatments, leading to an increased survival [312].

The principal miRNA delivery systems reviewed in this article, together with their main strengths and limitations, are summarized in Figure 2.

## 6. Concluding Remarks

miRNA-based therapies represent a very promising strategy to target T lymphocyte function, opening new possibilities for the treatment of immune-related diseases. This rapidly evolving field has led to an overwhelming number of pre-clinical and clinical studies in the last few years, as reviewed here, allowing the design of novel and more efficient therapies. Advances in the field include the combination of miRNA-carriers to improve the delivery and reduce off-target effects, such as coating with surface receptors, or co-delivery of complementary drugs. Moreover, further studies using different administration routes, dose-dependent efficacies and a better understanding of miRNA dysregulation in disease, will certainly allow the improvement of the use of miRNAs in nanomedicine.

## Figures and Tables

**Figure 1 ijms-24-00250-f001:**
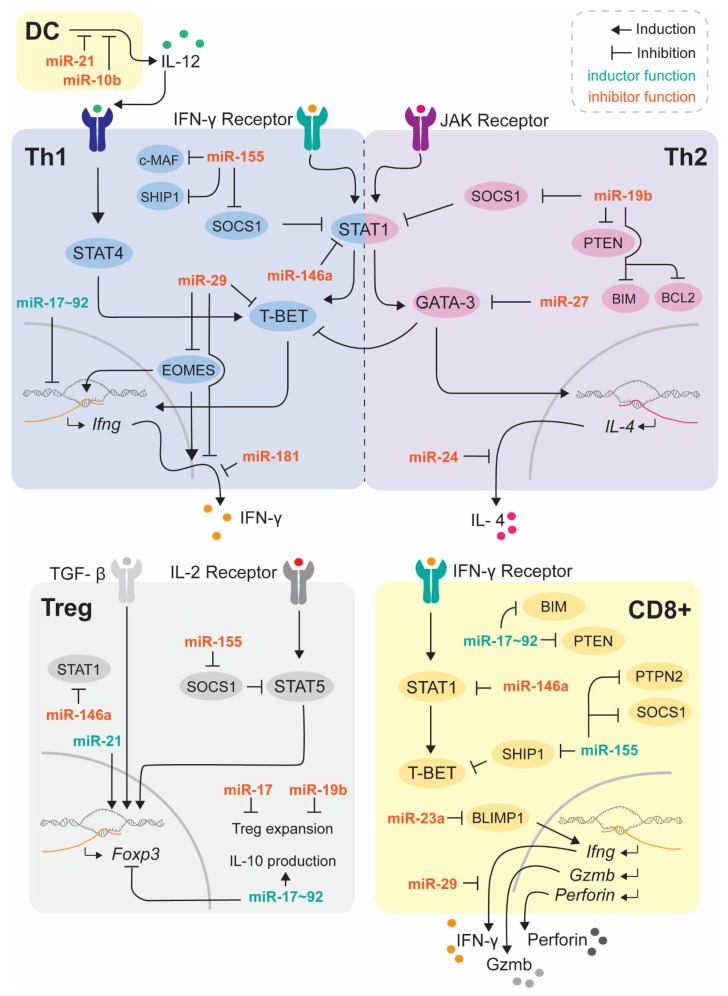
miRNA principal regulators of T cell function. This figure summarizes the best-established miRNAs involved in T cell function and polarization, that may be putative targets for T-cell immunotherapy.

**Figure 2 ijms-24-00250-f002:**
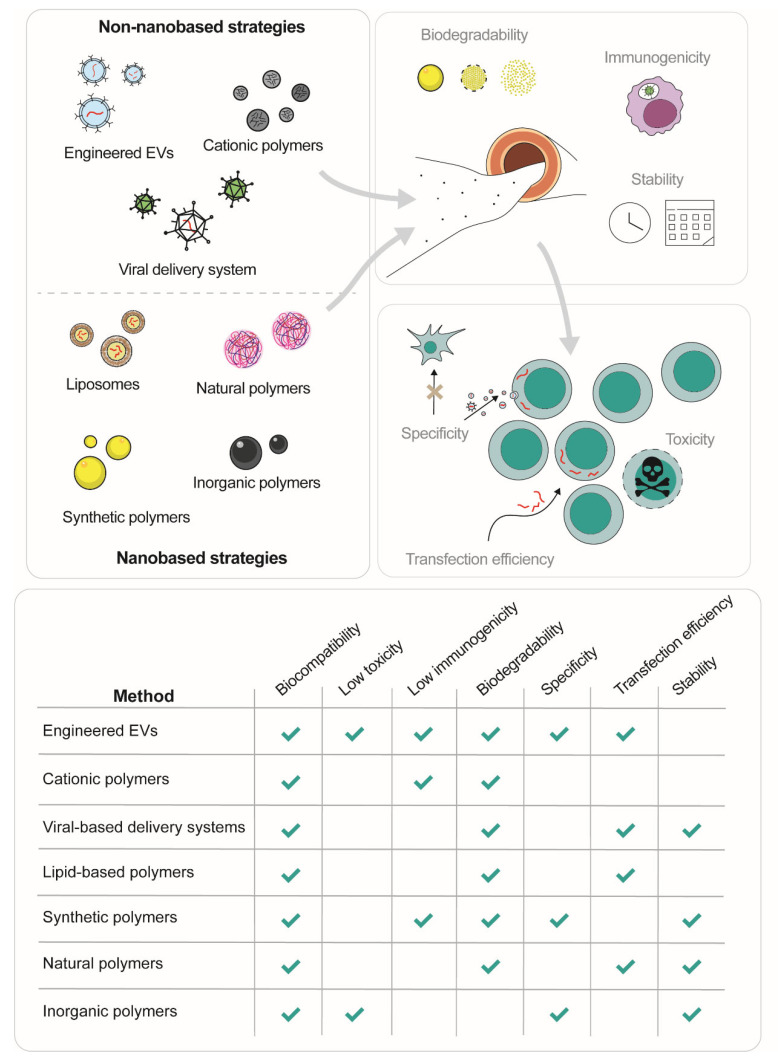
miRNA delivery systems. Summary of the main technologies available for miRNA delivery, summarizing their principal advantages.

**Table 1 ijms-24-00250-t001:** miRNA-based clinical trials.

	Drug	Clinical Trial Number	Type of Administration	Participants	Status	References
miR-124	ABX464 (Abivax S.A.)	NCT02792686	Oral dose	Healthy volunteers (24 participants)	Phase 1 completed March–July 2014	[174]
		NCT02731885	Oral dose	Healthy volunteers	Phase 1 completed September 2014–June 2015	[164]
		NCT02452242	Oral dose	Untreated HIV patients	Phase 2 completed January 2015–May 2016	[165]
		NCT02735863	Oral dose	HIV infected patients (30 participants)	Phase 2a completed May 2016–June 2017	[175]
		NCT02990325	Oral dose	HIV patients and healthy volunteers (36 participants)	Phase 1 and 2 completed March 2017–December 2018	[176]
		NCT03093259	Oral dose	Ulcerative colitis (32 participants)	Phase 2a completed October 2017–September 2018	-
		NCT05121714	Oral dose	Healthy volunteers (59 participants)	Phase 1 completed December 2017–May 2021	-
		NCT03368118	Oral dose	Ulcerative colitis (22 participants)	Phase 2a active January 2018–	-
		NCT03813199	Oral dose	Rheumatoid Arthritis (60 participants)	Phase 2a completed July 2019–April 2021	[177]
		NCT04049448	Oral dose	Rheumatoid Arthritis (40 participants)	Phase 2 active August 2019–	-
		NCT03760003	Oral dose	Ulcerative colitis (254 participants)	Phase 2b completed September 2019–April 2021	-
		NCT04023396	Oral dose	Ulcerative colitis (217 participants)	Phase 2b active January 2020–	-
		NCT04393038	Oral dose	SARS-CoV-2 infected (509 participants)	Phase 2 and 3 terminated July 2020–April 2021	-
miR-92a	MRG-110 (miRagen Therapeutics, Inc.)	NCT03603431	Intradermal injection	Healthy volunteers (42 participants)	Phase 1 completed April 2018–March 2019	[178]
miR-29	Remlarsen (MRG-201) (miRagen Therapeutics, Inc.)	NCT03601052	Intradermal injection	Keloid (14 participants)	Phase 2 completed June 2018–June 2020	-
miR-155	Cobomarsen (MRG106) (miRagen Therapeutics, Inc.)	NCT02580552	Subcutaneous and intratumoral injection	CTCL; MF; CLL; DLBCL; ATLL (66 participants)	Phase 1 completed February 2016–October 2020	-
		NCT03713320	Intravenous infusion	CTCL; MF (37 participants)	Phase 2 terminates April 2019–December 2020	-
		NCT03837457	Intravenous infusion	CTCL; MF (9 participants)	Phase 2 terminated October 2019–July 2020	-
miR-16	TargoMir (Asbestos Diseases Research Foundation)	NCT02369198	Intravenous infusion	MP; Mesothelioma; NSCLC (27 participants)	Phase 1 completed September 2014–January 2017	[166]
miR-34	MRX34 (Mirna Therapeutics, Inc.)	NCT01829971	Intravenous infusion	PLC; SCLC; L; M; MM RCC; NSCLC (155 participants)	Phase 1 terminated (five immune related serious adverse events) April 2013–May 2017	[164]
miR-122	Miravirsen (Santaris Pharma A/S)	NCT01646489	Subcutaneous injection	Hepatitis C Chronic Hepatitis C (5 participants)	Phase 1 completed June 2012–September 2012	-
		NCT01200420	Subcutaneous injection	Hepatitis C (38 participants)	Phase 2 completed September 2010–December 2011	[179]
		NCT02508090	Subcutaneous injection	Chronic hepatitis C (10 participants)	Phase 2 completed August 2013–January 2017	-
		NCT02452814	Subcutaneous injection	Chronic hepatitis C (8 participants)	Phase 2 completed May 2014–May 2017	-

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
