# Peer review of "MicroRNAs in T Cell-Immunotherapy"

_ijms, 2022, doi:10.3390/ijms24010250_

Round 1

Reviewer 1 Report

This is an interesting paper about the therapeutic potential of miRNAs and I consider that it meets the conditions to be published

Author Response

We thank the reviewer for his/her positive comments.

Reviewer 2 Report

microRNAs appeared to be critical regulators in various inflammatory and infectious diseases. In this manuscript, the authors investigated the functions of miRNAs on T cell activation and exhaustion and analyzed clinical reports of miRNAs’ effect on non-T cells and T cells in cancers and other immune-related diseases. Moreover, the authors inspected the studies on the clinical trials using miRNAs and their delivery methods elaborately. Considering a large body of miRNA studies has accumulated for years, the authors well-organized the evidence and indicated a promising future for miRNA applications in clinical therapies.

Here are some comments.

1. Line 110, as the authors mentioned, the expression of miR-146a in humans and mice was different. Are the results from mice work still applicable to human clinical therapies?

2. In section 3, 3.1.1 was a summary of miRNA functions on tumor cells and miRNAs indirectly affect immune response. However, 3.1.2 was talking about the direct effect of miRNAs on Treg cells. The authors might need to clarify the indirect and direct effects of miRNAs in the content or in the title.

3. In section 3.2, from lines 284 to 302, the mechanisms of miRNAs were discussed based on different diseases. But starting from line 303, the focus changed to general Treg cells not diseases anymore. The authors need to reorganize to make these paragraphs consistent.

4. Lines 336 to 340 mentioned HIV-related miRNAs. While, in line 299, the author already described HIV-related miRNAs. If this 3.2 section still emphasizes diseases, the authors could combine them together to line 299 to make the structure clearer. It is the same for line 346 which was talking about tumor-related cells. It could move to the cancer part.

5. For section 3.3, Since the title of the review is “T cell-immunotherapy”, this paragraph could be excluded from this manuscript.

6. The authors displayed many strategies to deliver miRNA efficiently. Is there any system to specifically deliver microRNAs to T cells or other immune cells?

Author Response

Reply: We thank the reviewer for his/her positive comments and have addressed all the concerns in the points below and modified the main text accordingly.

Here are some comments.

1. Line 110, as the authors mentioned, the expression of miR-146a in humans and mice was different. Are the results from mice work still applicable to human clinical therapies?

Reply: We agree with the reviewer, and acknowledge that this sentence may be misleading. We have decided to remove this comment on murine T cells, since controversial results have been found by different groups, and although it has been described that miR-146a can be downregulated in mice upon activation (as reviewed in Rodriguez-Galan et. al, 2018, ref.29), in a different study, also cited in this manuscript (Yang et. al, 2012, ref.28), miR-146a was found to be upregulated in both CD4+ and CD8+ T cells. Given this controversy, and the fact that the miRNAs with paramount function presented in this review are highly conserved, we rather focus on the functional data on miR-146a immune regulation capacity, without underlining data which may need further study.

2. In section 3, 3.1.1 was a summary of miRNA functions on tumor cells and miRNAs indirectly affect immune response. However, 3.1.2 was talking about the direct effect of miRNAs on Treg cells. The authors might need to clarify the indirect and direct effects of miRNAs in the content or in the title.

Reply: We agree with the reviewer that describing the regulation of the PD-1/PD-L1 axis in 3.1.1, we mainly focus on the regulation of PD-L1 expressed by immune and tumorigenic cells; while CTLA-4/B7 section in 3.1.2 mainly describes the regulation of CTLA-4. This is because, although the review focuses on T lymphocytes, those are the better characterized systems of regulation, even though its effect may be exerted indirectly. In order to further clarify this point we have specified this in the text and have included two references of PD-1 targeting by miRNAs (Yang. El al., Cancer Lett, 2018; Li et. al., Oncotarget, 2016).

3. In section 3.2, from lines 284 to 302, the mechanisms of miRNAs were discussed based on different diseases. But starting from line 303, the focus changed to general Treg cells not diseases anymore. The authors need to reorganize to make these paragraphs consistent.

Reply: In this section we do not aim to summarize the role of miRNAs in diseases, since very comprehensive reviews have addressed this question. We rather aim to focus on those immune-related diseases in which T cell function is compromised. Indeed, besides a little summary of immune diseases where miRNAs have been found to play crucial roles, we start with the Sjogren Syndrome, characterized by dysregulation of T lymphocytes, as it is one of the best characterized. We have now included the participation of T cells in the revised manuscript in order to make this point clear.

4. Lines 336 to 340 mentioned HIV-related miRNAs. While, in line 299, the author already described HIV-related miRNAs. If this 3.2 section still emphasizes diseases, the authors could combine them together to line 299 to make the structure clearer. It is the same for line 346 which was talking about tumor-related cells. It could move to the cancer part.

Reply: Following the reviewer´s advice we have modified the text, including the HIV references under the “infectious diseases” section. In the second case, since the cancer section focuses on immune checkpoints, we believe that this part is better organized as it is now, since we want to bring forward the studies on other immune cells that may help to improve T-cell based strategies.

5. For section 3.3, Since the title of the review is “T cell-immunotherapy”, this paragraph could be excluded from this manuscript.

Reply: Following the reviewer´s suggestion we have deleted section 3.3 in the revised form of the manuscript.

6. The authors displayed many strategies to deliver miRNA efficiently. Is there any system to specifically deliver microRNAs to T cells or other immune cells?

Reply: We thank the reviewer for pointing out this key issue. To date to our knowledge, there is no strategy of specific delivery to T cells or other immune cells other than include modifications using cell-specific surface molecules to promote specificity, as described in the text.

Reviewer 3 Report

The manuscript my Dosil et al is a comprehensive review on miRNAs as regulators of T-cell targeting immunotherapy. The authors describe inb a consice but through way the recent advances in miRNA delivery strategies, clinical trials and future perspectives in RNA interference strategies. I didn;t detect any flaws in the text which can be read pleasantly and I feel it will provide importnat information to the researchers in the field. The authors, should they feel doing so, could consider shorten the miRNA-delivery part which I feel covers at the end the take-home message. Other than that, I feel that the manuscript could be considered for publication without any further modifications or corrections.  

Author Response

We thank the reviewer for his/her positive comments. Considering the huge amount of bibliography on miRNA-delivery strategies, we believe that the information presented here might be useful for researchers in the field, although we understand that we may somehow cover the take-home message. For this reason, we prefer to include this information in the present form.